Forest floor temperature and greenness link significantly to canopy attributes in South Africa’s fragmented coastal forests

Pfeifer Marion marion.pfeifer@ncl.ac.uk 1
Boyle Michael J.W. 2
Dunning Stuart 3
Olivier Pieter I. 4
1 Modelling, Evidence & Policy Group, SNES, Newcastle University , Newcastle Upon Tyne , United Kingdom
2 Forest Ecology and Conservation Group, Silwood Park Campus, Imperial College London , Ascot , Berkshire , United Kingdom
3 School of Geography, Politics and Sociology, Newcastle University , Newcastle Upon Tyne , United Kingdom
4 M.A.P. Scientific Services , Pretoria , South Africa
Montagnani Leonardo
Electronic publication date: 2019 Jan 10
Publication date: 2019
Volume: 7
Electronic Location ID: e6190
Received 2018 Aug 30; Accepted 2018 Nov 30
Copyright: ©2019 Pfeifer et al.
Copyright year: 2019
Copyright holder: Pfeifer et al.
License: This is an open access article distributed under the terms of the Creative Commons Attribution License, which permits unrestricted use, distribution, reproduction and adaptation in any medium and for any purpose provided that it is properly attributed. For attribution, the original author(s), title, publication source (PeerJ) and either DOI or URL of the article must be cited.
License URL: https://creativecommons.org/licenses/by/4.0/

Keywords: Coastal forests, Forest edges, Eucalyptus plantations, Ground surface temperature, NDVI, Habitat microclimate, Fragmented landscapes, South Africa, Remote sensing, Thermal mapping

Funding: The Royal Society 2017 Round 1 - RG160847 The Royal Society supported the research under project grant number Research Grants 2017 Round 1 - RG160847. The funders had no role in study design, data collection and analysis, decision to publish, or preparation of the manuscript.

==============================
Tropical landscapes are changing rapidly due to changes in land use and land management. Being able to predict and monitor land use change impacts on species for conservation or food security concerns requires the use of habitat quality metrics, that are consistent, can be mapped using above-ground sensor data and are relevant for species performance. Here, we focus on ground surface temperature (Thermalground) and ground vegetation greenness (NDVIdown) as potentially suitable metrics of habitat quality. Both have been linked to species demography and community structure in the literature. We test whether they can be measured consistently from the ground and whether they can be up-scaled indirectly using canopy structure maps (Leaf Area Index, LAI, and Fractional vegetation cover, FCover) developed from Landsat remote sensing data. We measured Thermalground and NDVIdown across habitats differing in tree cover (natural grassland to forest edges to forests and tree plantations) in the human-modified coastal forested landscapes of Kwa-Zulua Natal, South Africa. We show that both metrics decline significantly with increasing canopy closure and leaf area, implying a potential pathway for upscaling both metrics using canopy structure maps derived using earth observation. Specifically, our findings suggest that opening forest canopies by 20% or decreasing forest canopy LAI by one unit would result in increases of Thermalground by 1.2 °C across the range of observations studied. NDVIdown appears to decline by 0.1 in response to an increase in canopy LAI by 1 unit and declines nonlinearly with canopy closure. Accounting for micro-scale variation in temperature and resources is seen as essential to improve biodiversity impact predictions. Our study suggests that mapping ground surface temperature and ground vegetation greenness utilising remotely sensed canopy cover maps could provide a useful tool for mapping habitat quality metrics that matter to species. However, this approach will be constrained by the predictive capacity of models used to map field-derived forest canopy attributes. Furthermore, sampling efforts are needed to capture spatial and temporal variation in Thermalground within and across days and seasons to validate the transferability of our findings. Finally, whilst our approach shows that surface temperature and ground vegetation greenness might be suitable habitat quality metric used in biodiversity monitoring, the next step requires that we map demographic traits of species of different threat status onto maps of these metrics in landscapes differing in disturbance and management histories. The derived understanding could then be exploited for targeted landscape restoration that benefits biodiversity conservation at the landscape scale.

Introduction

In increasing parts of the tropics, landscapes are experiencing anthropogenic loss and degradation of natural habitats, including primary forests, woodlands, and grasslands. The outcomes are landscape mosaics that comprise patches of natural habitat and regrowth, tree plantations and croplands of differing extents and management intensities. The subsequent erosion of biodiversity in these landscapes (Gibson et al., 2011) is an important global challenge for biodiversity conservation as well as climate change mitigation and food security (Godfray et al., 2010). Natural habitats deliver carbon storage, hydrology and microclimate regulation services and supply resources used in construction, energy generation and trade. They typically harbour more species (Gibson et al., 2011) and more threatened species (Pfeifer et al., 2017) compared to plantations or croplands.

However, the fate of individual species following land use and management changes are difficult to predict for at least three key reasons. First, we lack detailed knowledge on habitat and resource needs at landscape scales for many species, in particular in the tropics. Second, many models ignore that species perceive landscapes as continuous gradients in habitat quality and resource availability rather than as discrete categories of usable and unusable habitat (but see Pfeifer et al., 2017). Third, we lack common indicators characterising habitat quality, which so far has been measured through various structural attributes characterising habitats or impacts on species’ demographic traits (Chaplin-Kramer et al., 2015). Calls for standardised landscape metrics describing continuous variations in habitat quality to improve our ability to predict species responses to land use changes at the landscape scale are increasing (Mortelliti, Amori & Boitani, 2010; Pfeifer et al., 2017). Remote sensing has been advocated in the biodiversity literature as an ideal tool to obtain such metrics (Pettorelli et al., 2016), providing land surface data that are consistent, borderless, global and can be repeated across time. However, existing remote sensing derived metrics tend to focus on biomass or canopy vegetation productivity, which show inconclusive relationships with habitat attributes that matter to species.

In contrast, temperature variation within forest stands and vegetation growth on the forest floor have been linked to species performance and community structure. Temperature is a key driver for the growth, survival, and abundance of species, and in particular insects (Bale et al., 2002). Temperature can control species interactions (Rae et al., 2006; Hemmings & Andrew, 2017), community composition and diversity with reports of shifts in the abundance of insect species tracking fluctuations in ground surface temperature (Thermalground) (Retana & Cerdá, 2000). Climate effects on physiology and survival are also among the key hypothesised drivers of altitudinal migration among tropical bird (Barçante, Vale & Maria, 2017) and bat (Mcguire & Boyle, 2013) communities. The productivity of the ground vegetation, which in forested stands constitutes the herbaceous layer, is also likely to play an important ecological role for species (Gilliam, 2007). Ground vegetation growth, here measured as the normalised difference vegetation index of the ground vegetation (NDVIdown), mediates nutrient fluxes, produces short-lived aboveground biomass and provides resources to ground-dwelling organisms (Bromham et al., 1999; Stork & Grimbacher, 2006).

Forest canopy structure can be up-scaled from plots and mapped using remote sensing data (Hadi et al., 2017; Pfeifer et al., 2012; Pfeifer et al., 2016). For example, forest leaf area index, LAI, and fractional vegetation cover, FCover, measured in a forest landscape in Borneo showed mathematical relationships with satellite-derived spectral and texture information, which allowed mapping LAI and FCover for each pixel in the landscape (Pfeifer et al., 2016). Forest canopies are also mechanistically linked to both temperature and ground vegetation productivity through their effects on radiation fluxes within forest stands (Deardorff, 1978). Forest canopy structure modulates air temperature within forests (Hardwick et al., 2015; Von Arx, Dobbertin & Rebetez, 2012) and growth of the herbaceous layer (Royo et al., 2010; Shirima et al., 2015). Canopy cover insulates against temperature extremes and macroclimatic changes (Suggitt et al., 2011), while changes in canopy cover are expected to underlie temperature changes along gradients of forest degradation (Blonder et al., 2018), along forest edges (Didham & Lawton, 1999; Ewers & Banks-Leite, 2013), and with forest conversion to other land uses (Hardwick et al., 2015; Meijide et al., 2018). Using canopy structure maps with statistical models describing relationships between canopy structure and NDVIdown and Thermalground should hence provide an indirect way to upscale and map NDVIdown and Thermalground at landscape scale.

A key challenge for upscaling NDVIdown and Thermalground using canopy structure maps is to test whether general rules exist linking vegetation canopy structure to changes in Thermalground and NDVIdown. Here, we focus on Thermalground and NDVIdown as indicators of habitat quality in coastal human-modified landscapes of Kwa-Zulu Natal, South Africa. We test for habitat quality variation along gradients of tree cover from 0 (grassland) to >50% (natural coastal forests and Eucalyptus timber plantations) and measure across edges separating forests from other habitats. We test the following three main hypotheses: Thermalground increases with NDVIdown based on the mechanistic understanding that vegetation acts as heat and water reservoir (Matthews, 2005). NDVIdown declines with increasing canopy closure, leaf area and canopy greenness (NDVIup) due to decreases in light availability on the forest floor hampering ground vegetation growth. Thermalground declines with increasing FCover and LAI, as both not only act to filter out sun light but also prevent vertical mixing of air below the vegetation canopy (Hardwick et al., 2015). If our expectations on relationships between vegetation canopy structure and Thermalground as well as NDVIdown hold, this would provide a suitable way to develop upscaling algorithms for Thermalground and NDVIdown in tropical landscapes based on mapping canopy structure.

Material and Methods

Study area

Fieldwork was implemented in the KwaZulu-Natal province of South Africa between April 7th and April 22nd 2018 (Fig. 1). We concentrated our field campaigns on the coastal landscapes, which comprise fragments of coastal forests remaining after long-term historical and recent forest loss (Olivier, Van Aarde & Lombard, 2013), large tree plantations, small patches of natural grasslands and croplands. The forests represent the southernmost end of East African Tropical Coastal Forest which extends from tropical central Africa along the east African coast (Burgess, Clarke & Rodgers, 1998). They occur within the Maputaland-Pondoland-Albany biodiversity hotspot and the Maputaland Centre of Plant Endemism (Wyk & Smith, 2001), which support high levels of floristic endemism as well as a number of narrowly endemic species, including relict species. The climate of the study region is humid and sub-tropical with annual rainfall averaging 977 ± 149 mm year−1 and annual air temperature averaging 21.6 ± 1.0 °C (Worldclim data published by Fick & Hijmans, 2017).

Figure 1 Location of fieldwork in Kwa Zulu-Natal province, South Africa.

In April 2018, plots were set up in five main habitat types (i.e., natural grassland, open bushland, natural forests, edges of natural forests and timber plantations comprising Eucalytpus species) located along 11 linear transects: six transects across forest –edge –grassland (T01, T02, T07-T10), four transects across forest –edge –plantation (T03-T06) and one transect across forest-edge-grassland-bush (T11). Each transect comprised three plots (one per habitat type) except for T11, which comprised five plots. (A–C) A typical transect encompassing natural grassland (A), the forest edge (B) and the forest (C). (D) Zoom into the transects sampled at the northern boundary of Ngoye forest. The dense forest (C) is shown in dark red and is clearly delineated from the natural grassland (A). The map is a false colour composite using RapidEye imagery at 5 m pixel resolution. (E) Location of sample sites within the study region. In the 2018 plots, we sampled NDVIdown and Thermalground as well as and canopy LAI and FCover (the latter was not measured in grassland plots, as they did not feature trees). We increased sampling effort for the development of canopy structure maps from Landsat data (see Fig. 6). Canopy structure was measured at an additional 17 forest plots and 7 tree plantation plots in 2018. We also included an additional 52 forest plots and 11 needle leaved plantation plots measured in 2015 (E).

Data collection and processing

We sampled 35 plots for habitat quality attributes, i.e., measuring Thermalground using an Optris PI450 Thermal Imaging Camera (382 ×  288 Pixels, 29°  lens angle, 0.04 K thermal resolution, 7.5–13 µm spectral range) and NDVIdown as normalised difference vegetation index (NDVI) using a MAPIR camera (MAPIR; Peau Productions Inc., San Diego CA, US) with filters for the Red (660 nm) and Near-Infrared (850 nm) parts of the electromagnetic spectrum (16 MP sensor: 4,608 ×  3,456 Pixels, 82° HFOV (23 mm) f/2.8 Aperture. We measured LAI and FCover using hemispherical photography using a Canon 5D Mark II with a Sigma f2.8 fisheye lens and NDVIup using the MAPIR camera facing upwards.

The thermal camera used in this study has an internal sensor that delivers ambient temperature values by default and takes them into account during measurements. The camera also implements offset calibrations accounting for thermal drift in the thermal detector. This is implemented by a motor driven motion of a blackened metal piece (so called flag) in the front of the image sensor, so each image element is referenced with the same temperature. Weather did not change considerably over the field period (sunny with occasional clouds). We aimed to minimise the impact of time of day as confounding factor by acquiring most data between 10 am and 2 pm. This corresponds to peak day time temperatures and solar gain, as indicated by temperature recordings from nearby weather stations provided by National Climate Centre, South African Weather Service (Table S1). We also measured air temperature at 1 m height in the plots to test for its potential effects on Thermalground measurements (Senior et al., 2018). Whilst appropriate for the purpose of this study, our data can only be understood as temporal snapshots of ground surface temperature variations.

Sampling design

Approximate locations of transects were chosen prior to fieldwork based on our knowledge of the landscape and ease of accessibility. The final transect placement within the landscape was implemented using stratified random sampling. Plots were set up in five main habitat types (i.e., natural grassland, open bushland, natural forests, edges of natural forests and timber plantations comprising Eucalytpus species) located along linear transects (Fig. 2). We sampled a total of 11 transects (Table S1 ): six transects across natural forest—edge—natural grassland habitats (T01, T02, T07–T10), four transects stretching across natural forest—edge—Eucalyptus plantation habitats (T03–T06) and one transect stretching across natural forest—edge—natural grassland—bush habitats (T11). Forests comprised three different forest types: coastal lowland forest, scarp forest and peatland forest. Each transect comprised three plots (one per habitat type) except for T11, which comprised five plots (two bushland plots).

Figure 2 Sampling design for habitat attributes measured during fieldwork.

(A) Plot sampling . We acquired NDVIup, NDVIdown and Thermalground images at five sample points (S1–S5) located within a 5 m × 5 m subplot (pink shade) in the centre of each plot (20 m ×  20 m). We acquired fisheye images at 12 sample points (S1–S12) per plot. FCover is estimated from the fisheye images for each sample point, while LAI is estimated at plot level (one per plot). (B) Plot location. Plots were located along linear transects stretching from natural forests across the forest edge to other habitats. Darker shades indicate core areas of respective habitats.

We increased sampling effort for measurements of canopy structure, which were used to develop upscaling algorithms for predicting and mapping canopy structure in the landscape using remote sensing data: we sampled an additional 17 forest (including woodlands) and seven plantation (one: needle-leaved trees, six: fruit trees) plots during this field study. We additionally used hemispherical images acquired from 52 forests between 22∕06∕2015 and 05∕07∕2015 (Rolo et al., 2017) and from 11 needle-leaved tree plantations between 11∕08∕2015 and 13∕08∕2015 across the coastal landscape (Fig. 1).

Collection of habitat quality data

We acquired fisheye images in habitat quality plots of 20 m × 20 m dimensions following standard protocols (Pfeifer, 2015). In brief, at each plot, we took on average 12 high-resolution images through a digital camera and equipped with a hemispherical (fish-eye) lens with sampling points distributed within the plot (Fig. 2A) following the VALERI design (VAlidation of Land European Remote Sensing Instruments) and hence standard protocols developed for the Global LAI Project (Pfeifer, 2015). The camera was held at 1 m above ground, looking vertically upward from beneath the canopy. The levelled hemispherical photographs were acquired normal to a local horizontal datum, orienting the optical axis of the lens to local zenith.

We acquired NDVI images of the ground vegetation greenness (pointing the NDVI camera downward, NDVIdown) and of canopy greenness (pointing the NDVI camera upwards, NDVIup). We also acquired an image of the MAPIR ground target (i.e., targets of known reflectance values) at the start of each survey and we repeated this throughout the day if sky conditions changed. We acquired NDVIdown and NDVIup images at five sample points for each plot. Ground images were acquired using a 50 × 50 cm square made of metal rulers to delineate boundaries around each point (Fig. 2A). At each sampling point, we measured Thermalground as radiometric corrected values (saved as snapshots in *csv matrix format) pointing the thermal camera downwards, again using the 50 × 50 cm grids made from metal rulers to indicate boundaries around each sampling point. Thermalground data acquisition failed for the grassland plot at T02, which was excluded in relevant analyses.

Processing of habitat quality data

Leaf area index (sensu Plant Area Index, LAI, corrected for foliage clumping) was estimated from the fisheye images at plot level. Fractional vegetation cover (sensu canopy closure, FCover, in %) was estimated for each image and hence sampling point. Fisheye images were first processed using ‘in-house’ algorithms (available for download from the Global LAI Project website: https://globallai.wordpress.com/publications/) and the freeware CAN-EYE v 6.3.8 (Weiss & Baret, 2010), following standard steps (Pfeifer et al., 2012). In brief, the ‘in-house’ algorithms extracted the blue band from each fisheye image as the blue band achieves maximum contrast between leaf and sky. This is because absorption of leafy materials is maximal and sky scattering tends to be highest in that band (Jonckheere, Muys & Coppin, 2005a). The algorithm then applied the global Ridler & Calvard method (Ridler & Calvard, 1978) to the blue band extracted from each image for identifying the optimal brightness threshold that distinguishes vegetation from sky (Jonckheere, Muys & Coppin, 2005b). The algorithm then used the threshold derived for each image to create binary images of vegetation and sky from the blue band images, which were subsequently processed in the canopy analysis software CAN-EYE V6.3.8 (Weiss & Baret, 2010) limiting the field of view of the lens to values between 0 and 60° to avoid mixed pixels and thus misclassifications. Our approach thus estimates LAI as plant area index (PAI), which includes materials such as stems, trunks, branches, twigs, and plant reproductive parts (Breda, 2003). This approach is used in other indirect measurements of LAI and acknowledges that masking some parts of the plants to keep only the visible leaves is not correct and could lead to large underestimation of the actual LAI value as some leaves are present behind the stems, branches or trunk (Hardwick et al., 2015).

Each NDVI image was calibrated using the MAPIR Plugin (https://github.com/mapircamera/QGIS) within the spatial analysis software Quantum GIS v2.14.3. The plugin first loads the ground target image to find the calibration values. It then calibrates all survey images using those values. We subsequently processed calibrated NDVI images and thermal images to summary statistics using R statistical software (R Core Team, 2018) following steps outlined in Table 1. For some samples and plots, calibration produced non-sensible results for NDVIdown and those samples and plots were excluded from the relevant analyses (the forest plots in T02–T05 and T07 and the plantation plot in T03). The errors resulted from missing at-plot ground target images preventing us from calibrating images accurately.

Table 1 Processing steps involved when analysing NDVI and thermal imagery.

Metric	Processing steps	
NDVIup	1. Read *jpg image, which consists of three bands	
	2. Extract band 1 (RED) and band 3 (NIR)	
	3. Compute NDVI as (NIR-RED)/(NIR+RED)	
	4. Compute statistics: mean, median, minimum, maximum and standard deviation	
NDVIdown	1. Read *jpg image, which consists of three bands	
	2. Extract band 1 (RED) and band 3 (NIR)	
	3. Compute NDVI as (NIR-RED)/(NIR+RED)	
	4. Delineate region of interest in image (subset) using the clearly visible boundary	
	5. Compute statistics for subset: mean, median, minimum, maximum, number of pixels, and standard deviation	
Thermalground	1. Read *csv file and plot as matrix	
	2. Delineate region of interest on the displayed file using the clearly visible boundary using the same extent of 150 ×150 cells	
	3. Compute statistics for subset: mean, median, minimum, maximum, number of values and standard deviation	
Notes.

RED Red reflectance values

NIR Near-Infrared reflectance values

Figure 3 Habitat attributes and their variation within and across habitat types.

Sample size differed between habitats. (A) NDVIdown: 144 points sampled across 29 plots (Grassland: 35 points from seven plots, Bush: five points in 1 plots Edge: 54 points in 11 plots, Forest: 35 points in seven plots and Plantation: 15 points from three plots). (B) Thermalground: 169 points sampled across 34 plots (Grassland: 30 points from six plots, Bush: five points in one plot, Edge: 54 points in 11 plots, Forest: 60 points in 12 plots and Plantation: 20 points from four plots). (C) NDVIup 139 points sampled across 28 plots (Bush: five points in one plots, Edge: 54 points in 11 plots, Forest: 60 points in 12 plots and Plantation: 20 points from 4 plots). (D) LAI: data are only estimated at plot level and hence shown for one bush plot, 11 edge plots, seven woodland plots, 17 forest plots, and four plantation plots. (E) FCover : 528 points sampled across 40 plots (Bush: 15 points in one plots, Edge: 155 points in 11 plots, Woodland: 68 points in seven plots, Forest: 230 points in 17 plots and Plantation: 60 points from four plots).

Statistical analyses of habitat quality data

We used the R statistical software package version 3.5.1 for all statistical analyses (R Core Team, 2018). Statistics of quality data for habitats were summarised using boxplots generated on plot level data (LAI) and sample points (NDVIup, NDVIdown, FCover, Thermalground) (Fig. 3). We used pairwise Wilcoxon tests with Bonferroni adjustments to test for significant differences in in the summary statistics between habitat types sampled. We expected higher values of NDVIdown and Thermalground corresponding with lower values of FCover and LAI in open habitat types such as bushland and edges compared to forest interior and Eucalyptus plantations.

We aggregated FCover, NDVIdown, NDVIup, and Thermalground at plot level (LAI estimates were estimated at plot level only). Subsequently, grassland plots (no trees present) were assigned values of ‘0’ for forest canopy LAI and FCover and values of ‘−1’ for NDVIup. We developed linear and general additive models, the latter using the gam function in the mgcv package (Wood, 2017), to test for single predictor relationships between NDVIdown or Thermalground and NDVIup, FCover or LAI. We selected the final models based on explained variance (i.e., higher adjusted R-squared). If both models produced the same estimates of adjusted R-squared, we subsequently discuss findings from the simpler, linear model. Specifically, we tested for changes in mean, minimum and maximum of NDVIdown or Thermalground as a function of mean, minimum and maximum of NDVIup and FCover. We also tested for relationships between mean, minimum and maximum of NDVIdown and mean, minimum and maximum of Thermalground. We tested for changes in mean of NDVIdown or Thermalground as a function of LAI. To visualise key relationships, we used the ggplot2 package (Wickham, 2016) using the smoothing function and specifying the linear or general additive model dependencies. Finally, we modelled the effects of air temperature (Tair) measured at plot level on the relationship between canopy structure and Thermalground (Thermalground ∼ FCover * Tair).

Upscaling habitat quality data using Landsat imagery

We focused on Landsat Surface Reflectance Satellite Level-2 satellite product, i.e., satellite images that are freely available online, of high geospatial accuracy, and can be downloaded as surface reflectance data for comparisons over time and space (and hence are already corrected for atmospheric noise). We used two Landsat scenes acquired over the study landscape on June 4th in 2014. We used those images to upscale canopy structure data, specifically canopy leaf area index and fractional vegetation cover derived from the hemispherical images. We assumed that canopy structure did not change significantly since 2014 for the plots sampled, which is reasonable given the plots were located in woody ecosystems from little utilised shrubs to coastal forests, the majority being located within protected areas.

All raster analyses were implemented using the ‘raster’ package’ (Hijmans, 2017). We downloaded the Landsat 8 surface reflectance product (LASRC), derived from the Landsat 8 Operational Land Imagery data in each Landsat scene, from the USGS Earth Explorer after screening for clouds aiming to minimise cloud coverage over the landscape. We processed those reflectance data by setting pixels covered with clouds or haze to NA and only using pixels for which the pixel quality attributes indicated clear conditions (i.e., pixel quality attributes coded as 322, 386, 834, 898, or 1346) and excluding water bodies (i.e., pixel quality attributes coded as 324, 388, 836, 900, or 1348). We used Quantum GIS spatial software (QGIS Development Team, 2009) to mosaic the four scenes and cropped the extent of the raster mosaic to the study area using the ‘raster’ package (Hijmans, 2017). Reflectance (spectral intensity) measured in the red and near-infrared bands of the electromagnetic spectrum were used to compute three maps of vegetation greenness (i.e., the normalised difference vegetation index, NDVI (Tucker, 1979), the modified soil-adjusted vegetation index, MSAVI2 (Qi et al., 1994), and the two-band enhanced vegetation index, EVI2 (Jiang et al., 2008). We used the ‘glcm’ (Zvoleff, 2016) package in R statistical software to obtain indices of image textures for the red, near-infrared and shortwave infrared 1 bands in the Landsat imagery. We computed two indices to obtain texture information for a given pixel and its neighbourhood for each of these three bands: MEAN and DISSIMILARITY. This was implemented on a grey-level co-occurrence matrix with a 90 degree shift and 64 grey-levels and a window size of 3 × 3 pixels for each band (Pfeifer et al., 2016). We also computed mean and standard deviation maps of NDVI using the focal function in the ‘raster’ package specifying a moving window size of 8 pixels (Hijmans, 2017).

To map canopy structure, we combined canopy structure data acquired in 2015 with canopy structure data acquired during the fieldwork described here. We extracted reflectance, texture and vegetation greenness data onto each plot. The final dataset yielded a total of 109 plots measured for canopy LAI and FCover (with N = 79 forest plots, 7 woodland plots, 2 bush plots, and 21 plantation plots (including broad-leaved and needle leaved plantations). We developed Random Forest models linking spectral, texture and vegetation greenness data to canopy structure data after excluding predictor variables from the model that were highly inter-correlated (P > 0.6). We computed the models using the ‘randomForest’ package (Liaw & Wiener, 2002) in R (Table 2: final predictor variables included in predictive model). We subsequently used the final models to upscale plot measured canopy attributes to landscape scale excluding water bodies and any other NA regions from the resulting maps.

Table 2 Interrelationships between habitat attributes at plot level.

We compared general additive models (GAM) to linear models (LM) and reported on the variance explained by the model (adjusted R-squared, Radjust2), the Intercept and the estimate for the smoothing term (ST) or coefficients. Models were fit to data aggregated at the level of plots using the ‘average’ function. Model details are shown with ∗∗∗ indicating P < 0.001, ∗∗ indicating P < 0.01 and ∗ indicating P < 0.05. Canopy LAI and FCover were set to 0 for grassland plots and NDVI up was set to −1 acknowledging that there were no trees present. We tested whether including grassland plots in the modelling would significantly affect model outcomes (With grass; Without grass). Models chosen for reporting are highlighted by grey shaded cells.

		With grass	Without grass	
		GAM	LM	N	GAM	LM	N	
Thermalground ∼NDVIdow	Radjust2	0.21∗∗	0.21∗∗	28	0.30∗∗	0.31∗∗	22	
	Intercept	24.3∗∗∗	23.02∗∗∗		23.4∗∗∗	22.6∗∗∗		
	ST/Coefficient	1	3.97∗∗∗		1	3.06∗∗		
Thermalground∼FCover	Radjust2	0.51∗∗∗	0.51∗∗∗	34	0.31∗∗	0.30∗∗	28	
	Intercept	23.7∗∗∗	27.6∗∗∗		22.9∗∗∗	27.2∗∗∗		
	ST/Coefficient	1	−0.06∗∗∗		1.4	−0.06∗∗		
Thermalground∼LAI	Radjust2	0.52∗∗∗	0.52∗∗∗	34	0.35∗∗∗	0.35∗∗∗	28	
	Intercept	23.7∗∗∗	27.4∗∗∗		22.9∗∗∗	26.6∗∗∗		
	ST/Coefficient	1	−1.19∗∗∗		1	−1.0∗∗∗		
Thermalground∼NDVIup	Radjust2	0.38∗∗∗	0.30∗∗∗	34			28	
	Intercept	23.7∗∗∗	23.6∗∗∗		not significant			
	ST/Coefficient	1.98	−3.20∗∗∗					
NDVIdown∼FCover	Radjust2	0.28∗∗	0.24∗∗	29	0.33∗∗	0.33∗∗	22	
	Intercept	0.33∗∗∗	0.60∗∗∗		0.26∗∗∗	0.98∗∗∗		
	ST/Coefficient	1.61	−0.01∗∗		1	−0.01∗∗		
NDVIdown ∼LAI	Radjust2	0.20∗∗	0.20∗∗	29	0.18∗	0.18∗	22	
	Intercept	0.33∗∗∗	0.57∗∗∗		0.26∗∗∗	0.76∗∗		
	ST/Coefficient	1	−0.09∗∗		1	−0.14∗		
NDVIdown ∼NDVIup	Radjust2	0.39∗∗		29	0.51∗∗	0.36∗∗	22	
	Intercept	0.33∗∗	not significant		0.26∗∗∗	0.15∗		
	ST/Coefficient	2.38			2.84	0.75∗∗		

Results

Habitat quality variation between habitat types

Habitats differed significantly in habitat quality metrics measured in this study (Fig. 3). As expected, Thermalground decreased from grassland and bush plots to edge plots (pairwise Wilcoxon tests with Bonferroni adjustment, P < 0.05) and then to forest plots (P < 0.001). Thermalground for plantation plots was higher than for forest plots and lower than for grassland and bush plots (P < 0.01). NDVIdown showed different trends with habitat types compared to Thermalground and was higher in grassland, bush and plantation plots compared to edge and forest plots (P < 0.05). FCover was higher for edge, forest and woodland plots compared to plantation plots and lower in woodland versus forest plots (P < 0.05). NDVIup was lower in plantation plots compared to forest plots (P < 0.05) and was marginally higher in bush compared to edge plots (P = 0.052). Canopy LAI did not differ significantly between habitats.

Habitat quality attributes and their inter-relationships

Thermalground showed significant linear dependencies on canopy attributes (Tables 2 and 3, Figs. 4 and 5). Specifically, for each increase in FCover by 10%, Thermalground declined by 0.6 °C starting from 27.6 °C (Fig. 4B) and for each increase in LAI by 1 unit, Thermal ground declined by 1.2 °C decrease starting from 27.4 °C (Fig. 5B). The predictive capacity of the models improved when taking into account air temperature effects (Table 4). Furthermore, predicted effects indicate stronger buffering impacts of canopy closure under higher levels of air temperature (Table 4).

Table 3 Interrelationships between habitat attributes at plot level.

Model details shown with ∗∗∗ indicating P < 0.001, ∗∗ indicating P < 0.01 and ∗ indicating P < 0.05. Canopy LAI and FCover were set to 0 for grassland plots and NDVI up was set to −1 acknowledging that there were no trees present. Models were fit to data aggregated at the level of plots using the ‘min’ , ‘max’ and ‘mean’ functions respectively. LAI estimates were only derived at plot level. Bold numbers indicate that general additive models were the better fit compared to linear models. Response variables respectively were minimum, maximum and mean values of NDVI down and Thermal ground. Numbers are adjusted R-squared estimates for the statistical models fitted to the data. The type of the relationship (positive, negative, concave) did not change in comparison to models based on averages (see Table 2).

		NDVIdown	Thermalground	
		Min	Max	Mean	Min	Max	Mean	
NDVIup	Min	0.37∗∗	0.29∗	0.33∗∗	0.42∗∗∗	0.32∗∗	0.38∗∗∗	
	Max	0.39∗∗	0.45∗∗∗	0.47∗∗∗	0.44∗∗∗	0.34∗∗	0.40∗∗∗	
	Mean	0.40∗∗	0.35∗∗	0.39∗∗	0.43∗∗∗	0.32∗∗	0.38∗∗∗	
FCover	Min	ns	ns	ns	0.22∗∗	0.25∗∗	0.25∗∗	
	Max	0.24∗	0.26∗	0.28∗	0.50∗∗∗	0.39∗∗∗	0.46∗∗∗	
	Mean	0.28∗∗	0.24∗	0.28∗∗	0.50∗∗∗	0.47∗∗∗	0.51∗∗∗	
LAI	Mean			0.20∗∗			0.52∗∗∗	

Figure 4 Habitat attributes aggregated at plot scale and their inter-relationships.

We plotted habitat attributes for each sample point (grey dots) and aggregated at plot level for each habitat type. We used general additive and linear modelling (GAM and LM) to test whether Thermalground increases with NDVIdown, and for each whether they decline with attributes indicating increasing canopy closure. The graphs show the better fitting model for each relationship visualised as smoothed conditional means (grey line) with a 95 % confidence interval (grey shaded band). Model fits shown with ∗∗∗ indicating P < 0.001,∗∗ indicating P < 0.01 and ∗ indicating P < 0.05. Model details are shown in Table 2. (A) The decline of NDVIdown with increasing FCover followed a nonlinear pattern with the rate of decline increasing once canopy closure increases to more than 40–50% (GAM, N = 29, Radj2=0.28**). (B) Thermalground declined linearly with increasing canopy closure (LM, N = 34, Radj2=0.51∗∗∗). The model suggests that for each increase in canopy closure by 20%, Thermalground would decline by 1.2 °C. (C) NDVIdown displayed a non-linear convex dependency on canopy greenness (GAM, N = 29, Radj2=0.39∗∗. For values of NDVIup above 0.0, NDVIdown increased with increasing NDVIup, for values of NDVIup below 0.0, NDVIdown increased with declining canopy greenness. (D) Thermalground increased significantly with increasing NDVIdown (LM, N = 28, Radj2=0.21∗∗). We highlighted plots for which predictions of Thermalground were either much higher (plots c, d, e) or lower (plots b, e) than expected from the models. Grassland plots (red letters): Plot a was measured at midday during a hot and sunny day, whilst plot b was measured at Ngoye but a couple of hours earlier than plot a. Plot c measured at Enseleni Nature Reserve, featuring patches of high grass and bare soil, and plot d was located in an area with high grazing pressure at Sodwana Bay. Forest plot (green letter): Plot e was measured at Enseleni before midday.

Figure 5 Canopy LAI dependencies of Thermalground and NDVIdown.

We plotted habitat attributes for each plot and habitat type. We used general additive and linear modelling (GAM and LM) to test for significant relationships. The graphs show the better fitting model for each relationship visualised as smoothed conditional means (grey line) with a 95% confidence interval (grey shaded band). Model fits shown with ∗∗∗ indicating P < 0.001, ∗∗ indicating P < 0.01 and ∗ indicating P < 0.05. Model details are shown in Table 2. (A) NDVIdown declined significantly with increasing canopy leaf area (LM, N = 29, Radj2=0.20∗∗). (B) Thermalground declined significantly with increasing canopy leaf area (LM, N = 34, Radj2=0.52∗∗∗).

Table 4 Effects of air temperature on modelled relationships between canopy structure attributes and Thermalground.

We expanded the linear models, identified as best fit predicting the response of Thermal ground to either FCover or Thermalground (see Table 2) by including additional and interactive effects of air temperature (Tair). The predictive capacity of the models improved significantly as indicated by the adjusted R-squared (Radj2). We estimated intercept and slope based on the derived models for selected values of Tair (21, 23, 25, 27, and 29 °C. Model details are shown with ∗∗∗ indicating P < 0.001, ∗∗ indicating P < 0.01, ∗ indicating P < 0.05 and a indicating P < 0.10.

	Thermalground∼FCover*Tair	Thermalground∼LAI*Tair	
Intercept	2.074357	3.2155	
FCover or LAI	0.155010a	3.0012∗	
Tair	0.887836∗∗∗	0.8054∗∗∗	
FCover*Tair	−0.007211∗	−0.1424∗∗	
Radj2	0.73∗∗∗	0.78∗∗∗	
Tair = 21 °C	Intercept = 20.7, Slope = 0.00358	Intercept = 21.1, Slope = 0.0108	
Tair =23 °C	Intercept = 22.4, Slope = −0.01084	Intercept = 22.8, Slope = −0.274	
Tair= 25 °C	Intercept = 24.3, Slope = −0.02527	Intercept = 24.5, Slope = −0.5588	
Tair= 27 °C	Intercept = 26.0, Slope = −0.03969	Intercept = 26.2, Slope = −0.8436	
Tair= 29 °C	Intercept = 27.8, Slope = −0.05411	Intercept = 27.9, Slope = −1.1284	

NDVIdown showed significant nonlinear declines with increasing canopy closure and linear declines with increasing canopy leaf area (Tables 2 and 3, Figs. 4A and 5A). Specifically, for each increase in LAI by 1 unit, NDVIdown declined by 0.09 starting from 0.57 (Fig. 5A). NDVIdown showed non-linear convex relationships with NDVIup increasing with rising canopy greenness above an approximate threshold of NDVIup = 0.0 and increasing with declining canopy greenness (towards no trees) below that threshold (Fig. 4C).

Thermalground was significantly increased in grassland plots, which feature no trees. However, any clear relationship between Thermalground and NDVIup disappeared when excluding grasslands (Table 2). Finally, Thermalground and NDVIdown correlate with each other (Fig. 4D) and data suggest that for each increase in NDVIdown by 0.1, ground surface temperature increased by approximate 0.4 °C. This pattern held despite an expected influence of measurements acquired during different times of the day (i.e., measurement bias through sampling effect). The statistical model fitted to the relationship between Thermalground and NDVIdown underestimated Thermalground for five plots and overestimated it for eight plots. Plots whose Thermalground was higher than expected based on their NDVIdown include grassland plots measured around midday (a, c, d). Plots whose Thermalground was lower than expected from their NDVIdown include a grassland plot from Ngoye (b) and a forest plot from Enseleni (e), both measured earlier in the day (10 am, slightly cooler time of the day).

Mapping habitat quality using upscaling algorithms

The predictive capacity of the habitat quality mapping using Landsat reflectance data and derived indices was limited. Random Forest models explained 35% of the variability on the LAI data and 31% in the variability of FCover data (Table 5). We used the resulting LAI maps to map ground surface temperature and ground NDVI (Fig. 6) using models as detailed in the legends for Fig. 5.

Table 5 Final models used to predict canopy attributes from Landsat spectral reflectance data and derived spectral and texture indices based on N = 109 data points.

Random Forest models were computed with importance computation set to true and specifying 2,000 trees (models converged after 238 trees for predicting LAI and 36 trees for FCover). Predictor variables include: mean of shortwave- infrared (SWIRM) and near-infrared (NIRM) reflectances, dissimilarity of near-infrared (NIRD) and of shortwave- infrared reflectances (SWIRD), and the mean and sd of NDVI within a focal moving window of 8 pixels (NDVIMFocal and NDVISD). Predictor variables and their importance (with standard error in brackets) to the model predictions were ranked using the mean decrease in accuracy (%IncMSE) estimated based on random permutations using out-of-bag-Cross-Validation.

	LAI	FCover	
Final random forest model predictors	SWIRM+NIRM+NIRD+ SWIRD+NDVIMFocal	SWIRM+NDVIMFocal+NDVISD	
Variance explained	35.3%	30.5%	
Importance	NIRD	0.3064 (0.008)	NDVIMFocal	106.531 (2.083)	
NDVIMFocal	0.2816 (0.011)	NDVISD	63.572 (1.744)	
NIRM	0.2290 (0.009)	SWIRM	43.209 (1.648)	
SWIRD	0.1370 (0.008)			
SWIRM	0.1335 (0.007)			

Figure 6 Zoom into maps of LAI and Thermalground produced for the study region.

The zoom shows Ngoye forest, an ancient scarp forest fragment located around 10 km away from the coast. (A) We mapped LAI based on texture and spectral indices derived from Landsat 8 reflectance data (see Table 5 for model details). (B) We used the LAI maps with the statistical models described in Fig. 5 and Table 2 to map Thermalground. (C) The ancient forests are directly bordered by natural grasslands along large parts of the northern and southern edge of the forest (E, G). However, the landscape changes dramatically in the surroundings, with homesteads scattered in hilly areas (D) and sugarcane growing on small-holder farms and in lowland industrial plantations (H). Timber plantations (F) can be found along the coast and slightly further inland.

Discussion

Ground surface temperature and ground NDVI, both habitat attributes that have been linked to diversity, abundance and behaviour of animal species in different studies, are positively correlated. These habitat attributes are strongly linked to canopy structure attributes commonly mapped using remote sensing data. Specifically, opening canopies by about 20% or reducing canopy leaf area by 1 unit, would result in an increase of average temperatures on the ground by more than 1 °C. Ground vegetation greenness would decline nonlinearly as a function of increasing canopy closure.

These findings are not surprising based on the mechanistic understanding of radiation fluxes within vegetation layers (Deardorff, 1978; Best, 1998). They are likely to have important implications for microclimate within forests stands though, as tropical forests are changing rapidly with current global change drivers. Forest canopies are showing stress and die-back responses to repeated droughts, the latter acting in concert with other disturbance drivers to open forest canopies (Malhi et al., 2009). Disturbance has been shown to decrease canopy cover in the Amazon rainforests by 13% to 60% depending on disturbance intensity (FCover = 98% in intact forests, 85% in logged and lightly burned forests, 63% in heavily logged forests and 39% in heavily logged and burned forests) (Gerwing, 2002). Based on relationships derived in this study, such canopy cover declines would translate to increases in ground surface temperature of at least 3 °C in severely disturbed forests. Our results are thus in line with findings from a review on the thermal buffering capacity of forests, in which warming effects caused by land use change ranged from +1.1 °C in degraded forests to +2.7 °C in plantations, +6.2 ° C in pasture and +7.6 °C in cropland (Senior et al., 2017). Taking into account expectations on temperature extremes during the day (Blonder et al., 2018) may paint an even bleaker picture as stronger responses of temperature extremes to canopy structure changes would be expected (Ewers & Banks-Leite, 2013; Hardwick et al., 2015).

However, canopy cover has also been shown to recover more rapidly than forest biomass after logging (Pfeifer et al., 2016). And while remotely sensed canopy structure maps from rainforests in Malaysian Borneo suggest decreased canopy cover even ten years after logging and conversion to palm stands (i.e., FCover was 6 to 10% lower in logged forests and 25% lower in oil palms compared to primary forests (Pfeifer et al., 2016), these declines were not detectable at plot level in the field. These rapid recoveries may underlie recent observations of logged forests being able to retain the thermal buffering capacities of undisturbed forests (Senior et al., 2018) and further studies are warranted to investigate the spatial and temporal feedbacks between forest canopy degradation, forest canopy recovery and thermal environments on the forest floor over time.

Thermal cameras calculate the surface temperature on the basis of the emitted infrared radiation from an object, which itself depends on the temperature of the environment as well as on the radiation features of the surface material of the measuring object (Kastberger & Stachl, 2003). Adjusting emissivity values for each material encompassed in the field of view of the camera might increase accuracy of our temperature measurements. However, accounting for this emissivity is unlikely to significant alter the relationships we found due to the camera’s internal calibration. Furthermore, the surfaces we scanned are relatively similar and composites of soil, leaf litter, green herbaceous vegetation and grass in varying proportions. Whilst emissivity values for these materials vary with values of 0.93 (barren sandy soil), 0.96 (partial grass cover), and 0.98 (short grass and grassland) over the wavelength ranges our sensor is working in, composites of effective surface emissivity for soil–grass like vegetation surfaces was evaluated and estimated as mean of 0.98 (Humes et al., 1994).

Temperature changes induced by changes in forest canopies may increase or decrease growth rate or other fitness traits of species depending on the species’ optimum temperature, the temperature it currently experiences in its environment and the temperature it would experience after canopy changes. For example, herbivores and their growth rates respond more strongly to temperature than the growth rate of plants, while aboveground ectotherms show stronger thermal response of life-history traits than belowground ectotherms (Berg et al., 2010). The variability in responses is likely to result in changes in the network relationships between species (Berg et al., 2010) and thus species’ ecological roles within forest ecosystem processes (Ewers et al., 2015). Temperature effects are likely to be stronger for species in the tropics and in particular tropical ectotherms in forests (Deutsch et al., 2008; Potter, Arthur Woods & Pincebourde, 2013; Kaspari et al., 2015). Tropical insects are suggested to have narrow thermal tolerances and to track air temperatures close to their optimal temperature; they experience near-lethal temperatures faster than temperate insects and warming is expected to reduce their population fitness by up to 20% (Deutsch et al., 2008). Ground surface temperature increases are likely to act in concert with heat stress as the operational temperature of ectotherms is determined by both convection (the exchange of energy between body and air) and conductance (the direct transfer of energy between objects and surfaces) (Potter, Arthur Woods & Pincebourde, 2013).However, direct empirical evidence is rare and we suggest four steps following on from this pilot study:

First, to account for time of day and hence ambient temperature effects on ground surface temperature measures and to analyse interlinkages between both, ground surface temperature should be sampled together with air temperature throughout the course of the day and over several days, and if necessary seasons (e.g., dry versus wet seasons in the Afrotropics), which could be achieved using mini meteorological dataloggers and radiation sensors. This will allow us to capture variation in average temperature and temperature extremes as experienced by species in the forest understorey.

Second, studies could test and account for potential effects of emissivity variability. Assuming that the object’s temperature differs from ambient temperature this could be achieved by using a black flat object of known emissivity (0.98) as calibration target with three steps: (1) adjusting the emissivity of the infrared thermometer to 0.98, (2) acquisition of the temperature of the calibration target, (3) scanning the temperature of a directly adjacent area and modify the emissivity until the measured value corresponds to the temperature of the black surface.

Third, the tropical forest floor harbours a set of insect taxa believed to be distinct from the forest canopy (Stork & Grimbacher, 2006). Their abundance and distribution on the ground is probably linked to leaf litter (Rodgers & Kitching, 1998), ground vegetation , availability of host plants (Novotny et al., 2002) as well as microclimate (Schulze, Linsenmair & Fiedler, 2001). Thus, detailed species community and species behavioural studies looking at a gradient of surface temperatures along a gradient of canopy openness should be implemented for a range of taxonomic groups.

Fourth, habitat quality can have different meanings to species depending on the ecological scales they operate on and their interlinkages in trophic networks (Schulze, Linsenmair & Fiedler, 2001). For example, ground-dwelling insects feeding on detritus may be more affected by fine scale variation in temperature (Levesque, Fortin & Mauffette, 2002); (Lessard, Dunn & Sanders, 2009). In contrast, larger body-sized mobile bird species acquiring resources across larger spatial scales may be more affected by the availability of nesting places and the distribution of prey items in the landscape (Sekercioglu et al., 2007). Network analyses are rare, in particular for the tropics, but would allow us to determine whether changes in insects and ground vegetation due to microclimate changes are likely to propagate into changes in larger-body sized animal groups.

Conclusions

Accounting for micro-scale variation in temperature is seen as essential to improve biodiversity impact predictions using species distribution models (Suggitt et al., 2011). Thermal imaging of land surfaces can be implemented using unmanned aerial vehicles (Bellvert et al., 2014) and to some extent satellite sensors (Lee et al., 2015). However there are technical challenges in flying UAVs across many regions and changes in temperature below vegetation canopies (‘buffer effect’) would be difficult to detect. Our ground-based analyses show that canopy structure and below canopy ground surface temperatures show clear significant relationships, which could be exploited for mapping habitat quality metrics that matter to species. However, more work is needed to (1) reduce uncertainties in these relationships and (2) to improve canopy structure mapping (and hence subsequent ground surface temperature mapping) using remote sensing data. Our data seem to suggest that increasing sampling effort to capture spatial (along gradients of canopy cover and leaf area) and temporal (as function of day light, climate seasonality and climate extremes) variation in ground surface temperature would be beneficial to address the first point. The second point could be addressed by using data acquired at higher spatial resolution, as we have shown for forest degradation gradients in Borneo (Pfeifer et al., 2016) and by sampling canopy structure variation for a wide range of habitat types on the ground resolution. Either way, linking ground surface temperature (maps) to species demographic traits and abundance distributions in predictive biodiversity modelling (Pfeifer et al., 2017) would be the next essential step to truly determine the choice of ground surface temperatures as suitable habitat quality metric. This could then be exploited to design landscapes that maximise benefits from habitat restoration and management for biodiversity conservation and other ecosystem services.

Supplemental Information

Supplemental Information 1 Data used for analyses

Click here for additional data file.

Supplemental Information 2 Canopy structure estimates derived from hemispherical images at plot scale

Click here for additional data file.

Supplemental Information 3 Satellite derived data used in the Random Forest Models

Click here for additional data file.

Supplemental Information 4 Supporting Information Table S1

The table details the time of recording, the location of recording (plot name, geographic coordinates), the mean values for key habitat attributes and temperature recordings acquired by the weather station nearest to the plot between 9 am and 3 pm during the day of measurement.

Click here for additional data file.

Supplemental Information 5 Metadata for datasets

Click here for additional data file.

Supplemental Information 6 Overview on sensor data acquired during fieldwork

Click here for additional data file.

We acknowledge the logistic support of M.A.P. Scientific Services (https://www.mapss.co.za/, Pretoria, info@mapss.co.za) in the field. We thank the National Climate Centre of South African Weather Service for providing weather station data at hourly resolutions acquired at weather stations near the field plots during the field period.

Additional Information and Declarations

Competing Interests

Author Contributions

Data Availability

The authors declare there are no competing interests. Pieter Olivier is Co-Director of the start up company M.A.P. Scientific Services.

Marion Pfeifer and Pieter I. Olivier conceived and designed the experiments, performed the experiments, analyzed the data, prepared figures and/or tables, authored or reviewed drafts of the paper, approved the final draft.

Michael J.W. Boyle and Stuart Dunning analyzed the data, prepared figures and/or tables, authored or reviewed drafts of the paper, approved the final draft.

The following information was supplied regarding data availability:

Pfeifer, Marion (2018): Dataset_PeerJ_Canopy_Temperature_NDVI_2018. figshare. Fileset.

https://doi.org/10.6084/m9.figshare.7011692.v3.

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
