# Peer review of "Forest floor temperature and greenness link significantly to canopy attributes in South Africa’s fragmented coastal forests"

_PeerJ, doi:10.7717/peerj.6190_

## Round 0.1 · original submission · Major Revisions

Dear Marion Pfeifer,

We received two detailed, constructive reviews for your paper. Both reviewers acknowledge the relevance and of your study, and its overall good quality, but also they raise several questions which need to be answered in order to publish a more clear and reproducible study. The two reviewers are highly respected in this field and I recommend considering their indications carefully. I am confident that the overall quality of your paper will take advantage.

Please consider also my few indications:

In the abstract, please double check if in all the observed thermal and LAI ranges and in all the seasons a 1% of the variation in LAI can lead to 1°C difference in ground temperature. Also, please better define the mentioned ‘quality metrics’: a temperature increase cannot be simplistically considered as an improvement in habitat quality (suggesting that the optimum is at the edge of the range), but can be conversely a problem for the renovation of sensitive species. Increasing or decreasing the average temperature and thermal range certainly influences community structures, but presumably with an optimum at an intermediate point for the different biological communities.

Please consider also that the apparent temperature information given by thermal cameras is based on the emitted energy over the operational wavelength range of the detector, which can be influenced by emissivity properties of the object. A calibration of the data collected by the camera based on the different object emissivity is recommended, or at least the reader must be advised about this issue.

I am waiting for the revised version of your paper,

Sincerely
Leonardo Montagnani

Reviewer 1 ·

Basic reporting

Lines 136-: what species occurred in these plots? This information is entirely missing.

The raw data are supplied and, while I did not attempt to run the R code, generally look good. You do need to specify the version of R (and all packages) used to run this code, citing R correctly (see “citation()”).

Line 168: “leaf area index” and “plant area index” are not the same thing; clarify.
Line 208-: this is confusing, as the introduction (esp. 102-106) hasn’t prepared us much for this upscaling, and many details are unclear.
L. 272: “are not independent of each other”, i.e. “are correlated”?
L. 275: “Plots whose”?
L. 294-295: right. And this needs to be emphasized more, I think; it doesn’t take anything anyway from this study to note that almost everything here is predictable from such first principles
L. 313-324: interesting and well done
Figure 3: on the one hand, thanks for including this figure, as it’s confusing how sample size varies in this study. On the other, the experiment unit is the plot (yes?) not individual camera sample points. Be careful to use a common unit (i.e. the experimental unit)
Figure 4: how were these models chosen?
Figure 5: need to report p-values of these models in caption
Table 1: nicely done

Experimental design

The “two main questions” in lines 102- are not well defined. First, “interlinked” is vague. Second, “Can we statistically link…” is a poor framing that might open the authors to charges of ‘fishing’ for relationships they want to find a priori. Both of these should be framed as falsifiable hypotheses.

How were the plots (line 122) sites? Randomly? From remote sensing images? Clarify.

Validity of the findings

Line 203: good job on using Bonferroni adjustments

Line 130: this isn’t sufficient, as temperature variations in this 4+ hour window could seriously affect the results. What was the air temperature distribution during your measurements (e.g. from a nearby met station, if available)? How much temperature change occurred in this window, and what percentage of observations fell outside this window? How do you reconcile this with lines 275-278?

Additional comments

This study looks at the relationships between forest floor temperature, canopy greenness, and canopy leaf area across a number of field transects in coastal forests in South Africa. This is a potentially interesting subject, for reasons well laid out in the introduction. The ms is generally well written although there are many minor typographical and grammar errors the cumulatively make for distracted reading.

I’m not used to the PeerJ review format, but in general: there are many points of interest here, but there are many unclear points about the methods; the research questions are poorly (and potentially dangerously) defined; the study is actually quite limited and conclusions extremely tentative. See specific comments above and below.

1. Line 27 and throughout: “Random Forest” (capitalized) is the name of the algorithm
2. L. 81-82: but lines 71-72 say vegetation productivity show inconclusive relationships; does this conflict with text here?
3. L. 98: define NDVI
4. L. 110: repeated “and April”
5. L. 124: define MAPIR
6. L. 354-: “Our data seem to suggest that…” I agree with this careful language, given the limitations of the study design; this should be emphasized more in the abstract

Reviewer 2 ·

Basic reporting

*Please see separate doc for specific comments*

The language is generally clear and professional - a few confusing sentences but nothing major. References are occasionally missing or incorrect. Figures require more explanation in their legends, and some of the colour choices make them difficult to interpret. Raw data are available (metadata describing the columns and their units would be useful).

Experimental design

*Please see separate doc for specific comments*

Description of the different sites sampled was very confusing, with various different terms used and not all of them appearing in figures. Perhaps thinking about why these sites were chosen to answer the questions of this study would help the authors to settle on clearer terminology.

It's difficult to assess whether the statistical models were appropriate as they are not described in much detail within the ms. The fitted lines in figures seem to be a good overall fit, but were produced using ggplot rather than the models directly.

Validity of the findings

No comment.

Additional comments

This study addresses an important issue in ecology: how we can exploit remotely sensed data to map habitat characteristics that matter to species, at biologically-relevant spatial scales. While there is clearly more work to be done in upscaling these models, the findings presented here will certainly help to guide future research and on this basis I think it is an important piece of work. My main criticisms are in the description of what was done, rather than the study itself.

Annotated reviews are not available for download in order to protect the identity of reviewers who chose to remain anonymous.

---

## Round 0.2 · Minor Revisions

Dear Marion Pfeifer,
We received two reviews of the revised version of your manuscript. Both reviewers acknowledge the improvements done during the revision and suggest a few minor additional changes.

I would have also a suggestion from my side: when you are describing the climatic data of your study area (line 132), you could use the climatological standard normal values (1981-2010), for temperature and for precipitation, to gain more comparability with similar studies that may follow.

Sincerely,
Leonardo Montagnani

Reviewer 1 ·

Basic reporting

See below.

Experimental design

See below.

Validity of the findings

See below.

Additional comments

As before, this study looks at the relationships between forest floor temperature, canopy greenness, and canopy leaf area across a number of field transects in coastal forests in South Africa.

The authors have generally responded well to all my previous concerns, clarifying the methods and results in many spots and notably improving the introduction. As a result there is greater reproducibility and transparency, and the text is clearer and stronger throughout. Nice job! I have no further major concerns.

One potential issue: note that the “dat_all_PeerJ.csv” file doesn’t load correctly in many standard CSV readers on my system (Excel, etc). A programming-oriented text editor opened it without problem, but you may want to check the file encoding used.

Reviewer 2 ·

Basic reporting

Reads much better now. References have been added and corrected. Clear metadata included with the raw data. Unfortunately I wasn't able to find the revised figure legends; I hope that the authors addressed my concerns there as robustly as they have addressed my other comments. A few minor things below.

Line 18 – ‘above-ground’ shouldn’t have any spaces
Line 29 – repeat of word ‘that’
Line 36 – ‘field-derived’ shouldn’t have any spaces
Line 40 – 'requires that we map' not 'requires to map'
Line 76 – I’d delete the ‘And’ at the beginning of the sentence
Line 104 – great to see the Blonder et al. ref here, I was about to suggest it
Line 325 – should read ‘underestimated’ and ‘overestimated’ (change tense)
Line 327 – ‘whose’ instead of ‘those’
Line 343 – would probably get rid of ‘whilst’
Lines 397-422 – this is a very long para now. Maybe list the four suggestions as a bulleted list?

Experimental design

In the revised version I can see much more clearly what the authors did, and I am satisfied that their experimental design and analyses were robust and appropriate. Hypotheses are much clearer and research questions no longer 'dangerously' worded.

Line 162-163 – following up on comment 14 of reviewer 1, the authors have now specified that "approximate locations of transects were chosen prior to fieldwork based on our knowledge of the landscape and ease of accessibility". I’m curious how they were chosen exactly, within these approximate locations? Randomly?

Line 170 – why does T11 comprise five plots when there are only four habitat types (natural forest – edge - natural grassland – bush habitats)?

Validity of the findings

As before, I think the study meets a need in the literature. The authors are clear in outlining this as a pilot study, so while in some ways it is limited (as noted by reviewer 1), the authors acknowledge this and make several useful suggestions for further work.

Additional comments

Great job on revising this ms - it is a significant improvement. It's now a lot easier to read and follow what was done. The context and implications are also more clearly described in the Intro & Discussion, and the Figures look good too. Well done.

(Please note - the line numbers I quote above are referring to the document with track changes.)

---

## Round 0.3 · accepted · Accept

Dear Dr. Pfeifer,

I am pleased to inform you that your paper has been accepted for publication in Peerj. I wish to compliment you for the high quality of your research.

Sincerely
Leonardo Montagnani